# The Normal, the Radiosensitive, and the Ataxic in the Era of Precision Radiotherapy: A Narrative Review

**DOI:** 10.3390/cancers14246252

**Published:** 2022-12-19

**Authors:** Sandrine Pereira, Ester Orlandi, Sophie Deneuve, Amelia Barcellini, Agnieszka Chalaszczyk, Isabelle Behm-Ansmant, Liza Hettal, Tiziana Rancati, Guillaume Vogin, Juliette Thariat

**Affiliations:** 1Neolys Diagnostics, 67960 Entzheim, France; 2Inserm UMR 1296, Centre Léon Bérard, 69008 Lyon, France; 3Radiation Oncology Unit, Clinical Department, CNAO National Center for Oncological Hadrontherapy, 27100 Pavia, Italy; 4C.H.U Rouen, Service de Chirurgie ORL et Cervico-Faciale, 76000 Rouen, France; 5QuantIF-LITIS, University of Rouen, 76000 Rouen, France; 6Department of Internal Medicine and Medical Therapy, University of Pavia, 27100 Pavia, Italy; 7CNRS, UMR7365, IMOPA Université de Lorraine, 54000 Nancy, France; 8CNRS UMR168, Institut Curie, Université PSL, 75000 Paris, France; 9Data Science Unit, Department of Epidemiology and Data Science, Fondazione IRCCS Istituto Nazionale dei Tumori, 20133 Milano, Italy; 10Centre National de Radiothérapie du Grand-Duché du Luxembourg (Centre François Baclesse), 4366 Esch sur Alzette, Luxembourg; 11Department of Oncology, Luxembourg Institute of Health, 1445 Strassen, Luxembourg; 12Department of Radiation Oncology, Centre François Baclesse, ARCHADE, 14000 Caen, France; 13Laboratoire de Physique Corpusculaire IN2P3, ENSICAEN/CNRS UMR 6534, Normandie Université, 14000 Caen, France

**Keywords:** toxicity management, irradiation sensitivity, RT workflow

## Abstract

**Simple Summary:**

Despite the implementation of specific dose constraints on healthy tissues to maintain the theoretical risk of late toxicity below 5% five years after radiotherapy (RT), many patients experience “unusual” toxicity during their oncological follow-up. This narrative review describes the methods of individual radiation sensitivity (iRS) diagnosis, their impact on the RT workflow as well as initiatives to support clinical decision-making initiatives.

**Abstract:**

(1) Background: radiotherapy is a cornerstone of cancer treatment. When delivering a tumoricidal dose, the risk of severe late toxicities is usually kept below 5% using dose-volume constraints. However, individual radiation sensitivity (iRS) is responsible (with other technical factors) for unexpected toxicities after exposure to a dose that induces no toxicity in the general population. Diagnosing iRS before radiotherapy could avoid unnecessary toxicities in patients with a grossly normal phenotype. Thus, we reviewed iRS diagnostic data and their impact on decision-making processes and the RT workflow; (2) Methods: following a description of radiation toxicities, we conducted a critical review of the current state of the knowledge on individual determinants of cellular/tissue radiation; (3) Results: tremendous advances in technology now allow minimally-invasive genomic, epigenetic and functional testing and a better understanding of iRS. Ongoing large translational studies implement various tests and enriched NTCP models designed to improve the prediction of toxicities. iRS testing could better support informed radiotherapy decisions for individuals with a normal phenotype who experience unusual toxicities. Ethics of medical decisions with an accurate prediction of personalized radiotherapy’s risk/benefits and its health economics impact are at stake; (4) Conclusions: iRS testing represents a critical unmet need to design personalized radiotherapy protocols relying on extended NTCP models integrating iRS.

## 1. Introduction

More than 19 million new cancer cases are diagnosed worldwide each year [1]. Radiotherapy (RT), one of the cornerstones of cancer treatment, is involved in about 50% of cures, particularly in breast, prostate, cervix, head and neck, lung, and brain cancers, as well as sarcomas [2]. 

RT aims at deactivating tumor cells by using high-energy ionizing radiation to induce critical DNA breaks, particularly double-strand breaks (DSBs). Due to their physical properties, photons, which are commonly used in most RT facilities, can cause unwanted irradiation of the healthy tissues located within the RT field and surrounding the target, leading to specific toxicity. In clinical practice, we differentiate early expected toxicity transitorily, which has rapid effects such as dividing epithelial cells within 3 months from RT; and late toxicity, which can occur months to years after the completion of RT—an estimated 18% of adult survivors cope with permanent consequences [3,4]. The standard practice is the respect of specific dose constraints on healthy tissues maintains the theoretical risk of severe late side effects below 5% five years after RT. Despite these precautions, a significant proportion of patients still experience “unusual” toxicity at some point during their follow-up. “Unusual” toxicity is defined as a spectrum of unexpected tissue reactions with the following hallmarks: (i) Grade ≥ 2 occurs within the first 2 weeks of RT; (ii) Grade ≥ 3 lasts >4 weeks after the end of RT (early toxicity); (iii) Grade ≥ 3 occurs or persists >90 days after the end of RT (late toxicity).

Individual radiation sensitivity (iRS) has a substantial impact on the determinants of late RT toxicity, among other factors. iRS characterizes an individual’s tissue or cellular reaction to exposure to ionizing radiation. It is here used in the context of radiation doses that would normally induce no toxicity in the majority of the population classified as normal-responding individuals [5]. As with most biological functions, iRS follows a Gaussian curve characterized by an average value and a standard deviation (σ). At the left of this curve, the patients experience unusual severe tissue reactions, although their phenotype still appears grossly normal [6]. At its right, the patients may tolerate the recommended maximal doses (or even higher) on healthy tissues with an optimal tolerance. 

Here, we aim to review the iRS diagnosis methods and estimate the impact of an a priori knowledge of a patient’s iRS in the RT workflow, using extended normal tissue complication probability (NTCP) models and initiatives to support decision-making. 

## 2. Radiotherapy Toxicities

### 2.1. Overview

Recently, the clinical aspects of RT late toxicity have been thoroughly described—as simplistically summarized in Table 1—according to the irradiated healthy organs [7]. Subsequently, we will focus on RT-induced skin/soft tissue fibrosis—the most common, ubiquitous, and clinically evident late toxicity. Clinically injured skin may display atrophy, loss of elasticity, and severe induration that may limit local joint movement. Fibrosis is sometimes associated with telangiectasia, hair loss, and hyper/hypopigmentation and may lead to the loss of function of the irradiated organs [8]. 

### 2.2. Mechanisms of RT Toxicities

Among RT-induced effects, one of the most frequent is RT-induced fibrosis. It involves complex molecular, cellular, and tissue mechanisms currently investigated to generate hypotheses for tissue response biomarkers, predictive assays, or mitigating treatments.

The effects observed after RT may result from direct, intervention-related (e.g., molecular damage to DNA and other substrates, cell deaths) and/or systemic mechanisms (e.g., inflammation, immune response, vascularization) integrated at the temporal-spatial level. The dose delivered to a given structure, the tissue organization, the degree of cellular differentiation, and their proliferative and regenerative capacity impact the variability of tissue damage and clinical expression. 

In addition, inter-individual variability occurs in fibrosis severity after a given RT plan in a specific anatomic area. 

#### 2.2.1. Direct Effects: The DNA Damage Response

The nature of the radiation-induced phenomena depends on the time elapsed since the irradiation: the phenomena are generally physical, chemical, and then biological, evolving from the microscopic scale (atoms, molecules, cells) to the macroscopic scale (tissue or organ, individual, populations). The most critical consequences at the cellular level involve DNA [10]. The main DNA damage types induced by ionizing radiation are base damage (BD), Single strand Breaks (SSB), and Double Stand Breaks (DSB), which we will focus on as one of the critical effects sought by RT. Homologous recombination for the repair of DSB is mediated by the heptameric rings Rad52 that slide along DNA [11]. Once the ring is close to the breakout, the Rad51 protein is recruited and initiates the formation of a nucleofilament, which involves a multitude of proteins such as RPA proteins (Replication Protein A), XRCC2 and XRCC3 as well as BRCA1 and BRCA2 proteins [12]; for mammalian cells, non-homologous end joining (NHEJ) repair is the major DSB repair mechanism [13]; The Ku80 protein associates with the Ku70 protein and this heterodimer slides along the DNA until it reaches the level of the break where it recruits the DNA-PKcs protein: these three proteins form the DNA-PK complex. The repair proteins (Ligase 4 and XRCC4) are then recruited to ligate the 2 DNA ends. The cellular response to the repair of a DSB is coordinated by kinase-signaling cascades such as the ATM/CHEK2/p53 pathway activated by induced DNA DSBs. The ATM protein self-phosphorylates at the 1981 serine site, and this results in de-dimerization and activation. The phosphorylation of H2AX by pATM leads to the recruitment of several proteins to the damaged DNA site, including 53BP1, BRCA1, Chk1, and Chk2, and results in the cell cycle’s arrest and the activation of checkpoints before DNA repair. 

Finally, the irradiated cells either survive with accurate genetic information, survive with unrepaired lesions in more or less critical genes or die in their first generations. RT-induced death results from the various lethal contributions that may coexist, such as mitotic death, senescence, and apoptosis [14]. 

#### 2.2.2. Indirect Effects: Oxidative Stress Response Mechanisms

Ionizing radiations can indirectly create cellular stress through the production of reactive oxygen species (ROS) [15,16,17] that play a fundamental role in cell proliferation, motility, cycle, and apoptosis [18,19].

ROS may be generated in the extracellular compartments or within the mitochondria [20]. ROS are extremely toxic to tissues and can diffuse to the mitochondrial membrane [21]. p53 regulates the redox status [21]. 

Low ROS levels in the cellular compartments promote detoxification by the upregulation of reducing factors (i.e., nicotinamide adenine dinucleotide phosphate -NADPH- and glutathione) and the transcription of antioxidant enzymes (superoxide dismutases- SODs-, glutathione peroxidase 1, members of the sestrin gene family, and glutaminase 2) [22,23]. As the quota of ROS increases in the cell, p53 upregulates some genes (i.e., a p53-upregulated modulator of apoptosis (PUMA), p67phox, and p53-inducible genes) and suppresses nuclear factor-E2-related factor (Nrf2) to induce the transcription of antioxidant genes [24]. In cancer cells, p53 plays a crucial role in tumor cell death under increased intracellular levels of ROS. Cancer cells’ response to radiation-induced oxidative fueling can result in a hypoxic switch. Such an increase in the hypoxic proportion of cells, on the one hand, jeopardizes ROS production and, on the other hand, activates the hypoxia-inducible factor 1 (HIF1) in cancer cells [25,26].

#### 2.2.3. Systemic Mechanisms: Inflammatory and Remodeling Processes

The post-actinic inflammation begets a tissue remodeling process through a cross-talk between the repair and fibrogenesis pathways [27,28,29,30,31]. RT triggers the secretion of pro-fibrosing factors (i.e., TGFβ1) into the microenvironment causing the (trans)differentiation of mesenchymal, inflammatory, and epithelial cells into myofibroblasts which elicit an over-generation of extracellular matrix [29]. Moreover, the over-release of cytokines (TNFα, IL1, IL6) [32] and chemokines alter the management of oxidative stress within the irradiated tissues [33]. All these above-reported effects can result in a wide range of late toxicities. [7].

### 2.3. Risk Factors/Determinants of RT Toxicity

The following conditions may expose the patient to a higher probability/severity of toxicity for comparable dose/volume constraints as in the overall population not experiencing toxicities. Technical factors should be carefully examined when investigating the mechanisms behind putative radiation-induced toxicities. 

#### 2.3.1. Dosimetric Factors

Several therapeutic parameters may increase the risk of RT toxicity. 

There is a correlation between the probability of occurrence and the severity of tissue reactions on the one hand and the RT dose delivered in a specific body volume and in a certain time interval on the other. The total dose delivered is a major determinant of the outcome. Fraction size also has a relevant impact on late toxicities, with hypofractionation being more at risk of causing late damage than the conventional (1.8–2.2 Gy/day) fractionation schedules considering the same endpoint and the same volume of irradiated tissue [34]. Similarly, a high dose rate and a low interval between fractions (inferior to 6 h) may also increase the risk of late toxicity by saturating DNA repair mechanisms in healthy tissues [35].

All of these aspects should be modeled and integrated when planning RT, considering the specificity of dose-response for the different irradiated organs and tissues. Healthy organs at risk (OARs) surrounding tumor volumes are thus defined as structures to be spared more or less radically by the treatment when planning RT. Recommended doses for OARs are mainly derived from retrospective clinical experience accumulated over more than a century of practice [36,37] and correlate to dosimetric data to obtain Normal Tissue Complication Probability (NTCP) models [36,37,38,39]. More accurate guidelines may be expected from large databases prospectively collected and pooled with the standardized patient- or clinician-reported toxicities events and their corresponding dose maps [40].

The effects of the total dose, daily dose, dose rate, and treatment timing can be included in NTCP models. These models are predictive tools used in RT to estimate the risk of treatment toxicities. They convert relevant characteristics of the dose distribution in OARs into a predicted probability of achieving the outcome of interest. They can merge the combination of the dose distribution with patient, disease, and treatment characteristics [41]. NTCP models are currently used in clinical practice to optimize the planning of treatments and to guide the dose distribution in order to reach the optimal balance between Tumor Control Probability and the risk of toxicities. They can also assist the clinician in decision-making, e.g., selecting patients who would benefit the most from advanced RT techniques (rather than conventional radiotherapy), such as proton therapy, through a comparison of competing options for radiotherapy plans [42]. 

NTCP models can also include dose-modifying factors explicitly describing individual patients’ radiosensitivity, such as polygenic risk scores or results from biomarker assays [43,44,45,46]. These biologically-extended NTCP models can drive personalized decision-making and personalized optimization of treatments by setting goals on dose distributions that are tuned to the patient’s own genetics/biology [45,47]. Tucker et al. first proposed an NTCP model (mixture Lyman-Kutcher-Burman) for the prediction of grade≥3 lung pneumonitis, including the mean dose to the lung (defined as the total lung minus Gross Tumor Volume) and smoking status and 5 SNPs as modifying factors of the effective dose [48]. Five SNPs were included in the model (in TGFβ, TNFα, VEGF, XRCC1, and APEX1 genes). Other possible examples of such genetically extended NTPC models are those developed by Rancati et al. for predicting five urinary and rectal symptoms after radiotherapy for prostate cancer [47]. These models include clinical/treatment-related features (diabetes, presence of mild symptoms before radiotherapy, a previous transurethral resection of the prostate, a previous prostatectomy) and a polygenic risk score calculated from a cluster of SNPs identified within a validation study in the REQUITE population [43,44]. Deneuve et al. proposed the inclusion of results from the RADIODTECT© assay in NTCP models predicting acute, moderate, and severe oral mucositis and dysphagia after postoperative irradiation for head and neck cancers [46]. The gain in the discrimination power was evaluated in a pilot study. 

#### 2.3.2. Other Therapeutic Factors

It is acknowledged that a concomitant treatment during RT influences the RT’s effects [49]. Hormonal therapy increases the local post-actinic fibrosis and, indeed, an increased risk of skin and lung toxicity has been reported in women who underwent a concomitant tamoxifen treatment and greater loco-regional side effects in the case of androgen deprivation during pelvic RT [50]. A higher risk of RT’s local side effects is also described with the concomitant or sequential administration of anti-angiogenetics, target-therapy, PARP inhibitors, and Checkpoint inhibitors [51,52,53].

#### 2.3.3. Non-Genetic Clinical Factors

Tissue radiation sensitivity is a function of the developmental, self-renewal, and senescence dynamics of the organ [54]. Dysfunctions and growth disorders in irradiate organs are more specific to the pediatric population compared to adults [55,56]. At the other end of the age spectrum, the susceptibility to late toxicities in the elderly seems to involve not only a decline in the wound healing factor but also a shift in the mechanisms of radiation-induced cell death towards senescence, a deficit in DNA damage response, and an increase in oxidative stress and inflammatory response, interconnected with the frequently associated comorbidities [57]. Some acquired conditions (comorbidities) such as metabolic disorders including diabetes mellitus [58], hypertension [59,60,61], obesity (bolus effect) [62], infectious diseases including HIV infection [63], autoimmune or systemic inflammatory diseases including connective tissue and inflammatory bowel diseases [64,65,66] are associated with increased rates and severity of RT toxicities. The Charlson comorbidity index adjusted for age and the Adult Comorbidity Evaluation-27 (ACE-27) [67,68] showed different results when applied in different oncological settings, and its role remains controversial in predicting iRS. This increased individual toxicity risk may be due to immunodeficiency, microangiopathy, or the development of autoantibodies directed against DNA repair proteins [69]. It may be observed with conventional or high Linear Energy Transfer RT [70]. Habitus, such as tobacco consumption, also slightly increases the risk of RT toxicities [71].

#### 2.3.4. Radiosensitive Syndromes

A few infrequent hereditary diseases are characterized by a high iRS [72]. Such patients may suffer dramatic toxicity during/after RT (grades 4–5): these so-called hyper radiosensitive subjects carry inherited DNA damage repair deficiencies and a pathologic phenotype. They project to the far left outside of the Gaussian curve introduced earlier. 

A large number of autosomal recessive genetic syndromes associated with radiosensitivity are associated with DNA repair dysfunction, strongly suggesting that DSBs are the cause of radiation-induced cell death [72]. Homozygous ATM mutations causing ataxia-telangiectasia are associated with the strongest iRS in humans [73]. One could also cite LIG1, LIG4, NBS1, MRE11, FANC, BLM, XP, and CS mutations [74]. Other syndromes that do not directly link to DNA Damage Response genes can also be linked to significant iRS (e.g., Hutchinson Gilford Progeria syndrome, Huntington’s disease, Tuberous Sclerosis, Neurofibromatosis) [75,76,77,78]. These syndromes share a range of common clinical and biological characteristics, such as genomic instability, abnormal yields of chromosomal aberrations, immunodeficiency, and predisposition to cancer [5,79]. However, these extremely rare syndromes affect only a minor part of patients with RT-induced toxicities, as patients are usually denied RT based on their phenotype. 

### 2.4. Molecular Events Affecting iRS in Patients with a Normal Phenotype 

It has been estimated that more than 5–10% of patients suffering from high/moderate—but still “unusual”- toxicity during/after RT have none of the phenotypic alterations, risk factors, and pathogenic mutations involved in radiosensitive syndromes [74]. Deciphering the mechanisms underlying their iRS is, therefore, critical [80]. DNA damage response (DDR) and the oxidative stress response are the main cellular pathways involved in the iRS of patients affected by radiosensitizing genetic diseases. Some heterozygous carriers of mutated alleles of recessive radiosensitive syndromes, such as ATM, affecting 1% of the world’s population, may experience iRS [81]. However, the major phenotypic regulatory mechanism of gene expressions, such as Single Nucleotide Polymorphism (SNP), the epigenetic status of DNA and the influence of chromatin, changes in non-coding RNAs, telomere maintenance, and the irradiated cell’s microenvironment, including immune response, may mitigate iRS. It is currently unclear whether the cause of iRS in phenotypically normal patients is predominantly genetic or a consequence of non-genetic factors.

#### 2.4.1. Polymorphisms and Haplotypes

In the field of genome sequencing techniques and tools, radiogenomic studies have improved the knowledge of common genetic alterations associated with RT-induced toxicities. These studies have identified biomarkers through candidate gene studies, prediction models based on SNPs, or Genome-Wide Association Studies (GWAS) [82] (see Table 2 below).

Applied to the individual risk of RT-induced skin fibrosis, clinically significant gene expression patterns and regulators were associated with iRS. Several single nucleotide polymorphisms (SNP) were identified, most of them belonging to the DNA damage response (DDR) pathway -including ATM, a central kinase in the initiation of DDR [82]. Ho et al. demonstrated that the presence of any ATM variant in patients treated with RT for breast cancer led to a greater risk of developing RT-induced skin fibrosis [83]. The missense ATM SNP rs1801516 (c.5557G > A, p.Asp1853Asn) is currently the most promising SNP candidate in predicting RT-induced skin fibrosis, especially in breast and prostate cancer [84,85]. Interestingly, ATM IVS22–77 T>C/IVS48 + 238 C>G [86] and XRCC3 rs861539 [87] SNPs confer their respective radioprotective and radiosensitizing effect only at the heterozygous state. However, reaching the appropriate statistical power to clinically validate the isolated SNPs has always been challenging. In order to overcome this obstacle, Zschenker et al. proposed a composite risk score based on six risk alleles [88].

Mitochondrial haplogroups are also associated with iRS, the haplogroup H being radioprotective, while haplogroups J and U are associated with RT-induced skin fibrosis [89].

#### 2.4.2. Gene Expression and Alternative Splicing: An Invisible Burden That Matters

The use of skin transcriptome emerged in the early 2000s [92]. Quarmby et al. showed overexpression of PDGFB in skin fibroblasts from patients who experienced RT-induced skin fibrosis. Alsner et al. hypothesized that iRS determinants could lead to impaired radiation response of skin fibroblasts [90,93,94] and identified an in vitro transcriptomic signature for iRS in RT-induced skin fibrosis. Genes involved in RT-induced skin fibrosis were further associated with cellular functions involving the TGFβ pathway, extracellular matrix remodeling, apoptosis, proliferation, and ROS scavenging [94] (Table 3).

Forrester et al. identified a basal transcriptomic signature to distinguish patients prone to develop RT-induced skin fibrosis using fibroblasts from healthy skin. [95,96]. Overall, 1577 genes were expressed differentially between radiosensitive and radioresistant fibroblasts—associated with collagen metabolism gene ontology terms.

Phenotype-specific splicing events have also been highlighted: 152 genes were alternatively spliced between radiosensitive and radioresistant fibroblasts. They were associated with gene ontology terms linked to fibrosis’ pathogenesis, such as the integrin-mediated signaling pathway [97] and extracellular matrix disassembly. Only half of the alternatively spliced genes and 5% of differentially expressed genes are common, suggesting that independent regulatory mechanisms are involved.

#### 2.4.3. Role of Epigenetic Marks

Epigenetic modifications include histone modifications, such as acetylations and methylations, DNA methylation, particularly on the CpG island, non-coding RNAs, and three-dimensional chromatin organization [98]. Few studies have been conducted on skin RT-induced fibrosis epigenetic regulations, mainly on microRNAs [99]. Moreover, histone modifications, major determinants of gene expression, were shown to be involved in skin RT-induced fibrosis as treatment with phenylbutyrate, a histone deacetylase inhibitor, was correlated with suppression of the aberrant expression of RT-induced TGFβ and TNFα expression, subsequently reducing skin RT-induced fibrosis [100]. Weigel et al. conducted a whole genome epigenetic analysis on radiosensitive and radioresistant skin fibroblasts [101]: 12,968 differentially methylated CpG islands covering 9060 genes were identified. Interestingly, a gene ontology analysis revealed that these genes were associated with transcription regulation functions, such as regulation of transcription by RNA polymerase II and mRNA splicing via the spliceosome, and with fibrosis-linked functions, such as extracellular matrix organization or the integrin-mediated signaling pathway.

## 3. Predictive/Prognostic iRS Biomarker Research

At the molecular level, oxidative stress, DNA and biomolecule damage, DNA repair, cell death, and local and systemic reactions are involved in the occurrence of RT toxicities. The association of genes or their products or regulators in these different pathways was studied as a potential biomarker for toxicity prediction (Figure 1).

The challenge is to implement a reproducible assay with high sensibility and sensitivity [102]. Alternative, more high-throughput approaches are being developed for routine practice. Several approaches are proposed [103], including (but not limited to) RT-induced lymphocyte apoptosis [104], quantification of radiation-induced pATM [105], TGFβ1 genetic variation [106], and spontaneous transcriptomic signature targeting RNA involved in RT-induced fibrosis.

### 3.1. Systemic and Radiological Biomarkers

Predictive biomarkers of iRS [107] can be plasmatic such as pro-inflammatory cytokines (i.e., IL1α, IL6, IL8), growth factors (i.e., TGFβ1) and blood cell levels (i.e., hemoglobin, neutrophils, neutrophil-to-lymphocyte ratio (NLR), systemic immune-inflammation index) as well as genetic (i.e., single nucleotide polymorphisms- SNPs). Their level may change under RT, and they may serve as direct markers of response to RT.

In the radiomic approach, normal tissue imaging in response to RT or dose distribution [108] may predict toxicity in the head and neck (xerostomia) [109] and lung (pneumonitis) [110] cancers. A possible association between imaging features and gene expression was also proposed and is currently being investigated. The possibility of an imaging biomarker giving an insight into genetics/genomic/transcriptomics is highly tempting, as it would be non-invasive, interrogating large tissue volumes and allowing a possible longitudinal evolution. Yet, one should acknowledge the big difference in spatial scales between imaging and genetics [111].

### 3.2. Genetic Assays

Numerous studies tried to identify mutations and polymorphisms in patients with different types of cancers. Single nucleotide polymorphism (SNP) studies are limited to a candidate gene approach, with a tissue specificity for each genetic determinant and “linkage disequilibrium” for which some SNPs can catch most of a regional genetic variation. Even if most studies have produced statistically significant results, the models were often not applied to validation cohorts (Table 4). The most significant progress in these areas has been made in identifying specific SNPs associated with late toxicities in breast and prostate cancer via the European REQUITE study [40,43,44]. The French study PROUST is in progress and centralizes data and blood samples to validate these predictive markers of individual RT-induced toxicity [112].

### 3.3. ATM-Based Assays

Sensitivity to RT can be associated with rare cancer-prone syndromes [6]. ATM protein kinase is a key component in the cellular response to DNA double-strand breaks. Furthermore, a mechanistic model based on the radiation-induced-nucleoshuttling of the ATM protein (RIANS) was also used to explain iRS of genetic syndromes caused by mutations in cytoplasmic proteins such as Huntington’s disease, neurofibromatosis and Tuberous Sclerosis syndrome [76,77]. Based on these data, two assays have been developed and used fibroblasts or lymphocytes derived from patients [46,105,120,121,122]. In the fibroblasts of 117 cancer patients, a quantitative correlation was found between the maximal number of nuclear pATM foci assessed by immunofluorescence in the first-hour post-irradiation and the severity of the RT-induced toxicities assessed by the Common Terminology Criteria for Adverse Events (CTCAE) severity grades [105,120]. Since the first predictive assay based on the RIANS model required immunofluorescence and cellular amplification, an assay based on the quantification of the nuclear pATM forms with the ELISA technique was developed [121]. In order to simplify the test; the ELISA assay was adapted to lymphocytes extracted from a blood tube. The validation of this ELISA test on lymphocytes was then carried out on a cohort of 150 patients with different types of cancer, for which we obtained a discrimination power of 0.77 (Area Under the ROC) [46], as well as on the second cohort of 40 patients with Head and Neck cancers. A study on a French cohort of 36 HNSCC patients has demonstrated that the ELISA pATM assay can be combined with NTCP dosimetric models with AUCs ranging from 0.72 to 0.80 [122].

### 3.4. Apoptotic Assays

Apoptosis is the most crucial pathway of programmed cell death. Ionizing radiation can induce cell apoptosis phenomena linked to DDR with activation in a p53-dependent manner, the release of cytochrome C of mitochondria, and apoptosome formation [123,124,125]. On the other hand, radiation-induced cell apoptosis can also be the consequence of the activation of the extrinsic apoptotic pathway (depending on death receptors) [126] or the apoptotic stress pathway membrane (depending on the activation of a cascade of mitogen-activated protein (MAP) kinases and caspases leading to the fragmentation of nuclear DNA) [127]. Previously, it has been shown that apoptosis is linked to iRS in lymphocytes [104]. The radio-induced lymphocyte apoptosis (RILA) quantification is an assay that evaluates the quantity of apoptotic peripheral blood lymphocytes after ionizing radiation exposure (with 2 or 8 Gy) [128,129,130]. The assay was correlated with late toxicity in several studies in prostate and breast cancers [131,132,133] and with radiation-induced sarcoma in women previously irradiated for breast cancers [134]. RILA was recently shown to be associated with acute pain in breast cancers, thanks to the European REQUITE study [135].

### 3.5. 8-Oxo-Guanine Assays

The oxidative stress response has been described for a long time as being involved in determining the response to irradiation. Some assays to predict radiation toxicities have focused on the quantification of enzymes involved in DNA repair mechanisms of the base, nucleotide, single DNA, or double DNA strand breaks: one such first study on 38 patients with head and neck cancer showed that the dysfunction of these pathways might be associated with some risk of toxicity depending on the treatment’s outcome [136].

### 3.6. Pros and Cons of Available Methods

The last 20 years offered exponential advances in genomic technology, as nowadays, SNP or GWAS studies can be performed directly from a blood sample. This technology is on the edge of offering new tools for targeted screening in high-risk individuals, but more research is needed if GWAS is to pay off the investment required (see Table 2, Table 3 and Table 4): nowadays, GWAS studies require large cohorts to identify SNPs [117,129], and in most cases, these results are not reproducible because those other factors than iRS impact the risk of toxicity such as radiation dose, treatment (chemotherapy, surgery), age, and comorbidities, which highlights the need to collect and include multiple variables in studies [137]. The best-documented functional tests analyze, at the molecular level, phenomena directly related to irradiation, DNA Damage Repair, or Apoptosis. However, unlike the RILA test, which has been tested prospectively on large cohorts [80,135], the RIANS assay based on fibroblasts and blood needs larger prospective studies to be validated [80].

## 4. Practical Consequences of a Priori Knowledge of iRS for Precision Radiotherapy and Patient Selection

### 4.1. Screening of Individual Radiation Sensitivity

Systematic screening of radiosensitivity in patients scheduled to undergo radiotherapy is undoubtedly a way to further reduce the morbidity of radiotherapy in a personalized fashion. The development of high-performance iRS diagnostic tests should make this strategy feasible in routine practice. A prerequisite for this is that such testing has to be integrated into treatment costs. Pre-therapeutic radiosensitivity testing would then be used to adapt cancer treatment a priori.

Currently, iRS screening tests have ~90% accuracy. Therefore, there is no clinical obstacle to implementing iRS screening in practice. Therapeutic modifications could then mainly consist of treatment de-escalation with a reduction in the administered doses or the irradiation of more limited, selected volumes. Strategies based on iRS testing have to rely on evidence-based medicine, such as non-inferiority clinical trials, to ensure that tumor control is at least equivalent to a personalized treatment as with standard treatment and that its associated toxicity rates are indeed lower.

Another prerequisite for the implementation in routine practice is that a standard technique for iRS screening should be recognized internationally as there are currently multiple and non-consensual iRS approaches.

The recommendations of the American Society for Radiation Oncology [138] are currently limited to the use of genetic testing in patients that are known to be at risk of toxicities linked to excessive radiation. Apart from ATM mutations, the authors state that “radiation therapy recommendations should not be altered based on heterozygous mutations identified on genetic screening studies when radiation represents a clinical benefit.”

Therefore, the question of patient management according to their iRS remains open, as “unusual” toxicity events may occur even in patients without any identified genetic mutation. Although there is no international guidelines, physicians facing toxicities that are not explainable by technical factors or comorbidities in a given patient may be willing to investigate their iRS to adapt anticancer and supportive care management. It is estimated that about 90% (unpublished data) of pATM testing in France falls in this category. Given that the probability of iRS in a specific patient is high, physicians could propose customized monitoring of toxicities during further radiotherapy courses based on an accurate testing method.

Knowledge of the prediction of adverse events after RT has improved in the last years [46,85,105,106,107,108,109,110,111,112,113,114,115,116,117,118,120,121,122,128,129,130,131,132,133,134,135,136], so much so that some new avenues may be considered in the therapeutic management of patients. Early diagnosis of toxicities is possible when subclinical biological or radiological signs appear and are associated with unusual early or late toxicity events. Imaging methods are being implemented with that intent—such as deep learning and radiomics on high-throughput imaging data and high-resolution images such as functional MRI [139,140]. Biomarkers, such as TGF-β1 and IL-6, have been used in predicting radiation-induced lung disease [141].

In another field, mobile applications can be worth the remote monitoring of outpatients to improve the collection of toxicity data and the patients’ compliance and adherence to supportive treatments, which may help to avoid higher rates or the increased severity of acute and late toxicities [142,143].

### 4.2. Possible Therapeutic Adaptations in Patients Screened “iRS Positive”

Various physical, technological and biological approaches may be proposed to improve the profile of tolerance to RT in patients with normal phenotypes identified as radiosensitive (Figure 2). It may be challenging to estimate a risk/benefit ratio of radiotherapy in a given patient with a normal phenotype but with unusual toxicity for which iRS is suspected but not tested. iRS testing could provide clinically relevant support to decision-making for further estimates of the clinical benefit of RT, to be discussed in a multidisciplinary management setting. Depending on the level of risk of local recurrence, an alternative to RT may be considered with additional or more radical alternative treatments. The risk-benefit is to be discussed with the patient (see Clinical decision-making process for RT): for instance, mastectomy with immediate reconstruction may sometimes be preferred to lumpectomy with RT for in situ breast cancer management [144]. On the other hand, it is essential to consider that despite unavoidable complications, irradiation may still be required to achieve a cure in advanced stages due to the high probability of locoregional relapse [145]. Each decision should be made by an experienced multidisciplinary team and agreed upon with the patient who, if identified as moderately radiosensitive (in contrast to hyper-radiosensitive patients for which radiotherapy is to be denied), should fully understand the medical tradeoffs.

When RT avoidance represents suboptimal therapy, technical RT adaptations should be proposed to minimize toxicity.

#### 4.2.1. Tissue Sparing Techniques

Reducing the tissue volume irradiated at intermediate/high doses may help to reduce the incidence of toxicities—in 2022, over 90% of routine RT treatments rely on photon-based RT.

Photon-based techniques

Conformal irradiation, i.e., “conventional” tridimensional radiotherapy, using a linear electron accelerator producing “high energy” megavoltage-photons has gradually been replaced by Intensity-Modulated Radiation Therapy (IMRT) in many tumor sites treated with curative intent. Stereotactic Body Radiation Therapy (SBRT) has also been massively implemented in routine practice since the development of extracranial SBRT equipment with in-room imaging and the versatility of linear accelerators. SBRT is used in curative indications and increasingly also for metastases, sometimes with repeated courses. A shared trait of both IMRT and SBRT is the use of photons. The “unfavorable” dose deposit of photons, i.e., with a significant amount of dose under the skin continuing their course in tissues behind the tumor, has somewhat been compensated by the technological advances, i.e., with the use of many (coplanar or non-coplanar) beams and sophisticated photon fluence modulation using dynamic multileaf collimators. Manipulation of photon beams have resulted in the ability to deliver very steep dose gradients with better conformality. Clinically, these advances have translated into reduced volumes of organs at risk of receiving a high dose and, subsequently, into lower rates of clinically relevant toxicities. Control for setup uncertainties and tumor or organ tissue volumes (inter-fractions and intra-fractions) is exploited to reduce radiotherapy margins; this is the field of image-guided (IGRT) and adaptive radiotherapy (ART).

However, this better conformation has not fully eliminated the issue of low to intermediate doses to healthy tissues [146], and partial irradiation may be proposed in specific situations. Such a reduction in the volume of tissue that is irradiated at high doses is actively investigated in several clinical indications, including breast cancer or bladder cancer [147]. In contrast with schemes previously recommended for their protective radiobiology on normal tissues, current trends are to use fewer higher doses per fraction (hypofractionation) by highly conformal SBRT, and excellent tolerance profiles are being reported. Hypofractionated SBRT may modify repair mechanisms in the tumor area with low total doses in fewer fractions than IMRT in normal tissues.

Tissue volume-sparing with other forms of radiotherapy than IMRT or SBRT

As with -MV photon-based IMRT and SBRT, brachytherapy and hadrontherapy can be used to reduce the irradiated volumes [148]. Hadrontherapy, i.e., protons, helium ions, and carbon ions (also called heavy ions), has favorable spatial dose distribution compared with photons. The loss of kinetic energy at the end of the charged particle range produces a massive local dose deposit (the Bragg Peak, BP) with no dose located behind. The Bragg Peak is further spread out (SOBP) to irradiate the entire tumor thickness. Subsequently, irradiated volumes are roughly twice smaller with protons than with photons. Carbon ions also show less lateral scattering than protons, but they fragment into lighter ion species behind the Bragg peak. These lighter ions and their cross sections are insufficiently characterized, and these high-LET secondary particles can be responsible for toxicities [149,150,151].

Concerning protons, major technological advances have been made in the last 15 years, which have favored the expansion of protontherapy (PT) centers worldwide. Production of secondary neutrons has been reduced with active (pencil beam scanning) delivery (compared to passive PT).

It is also important to note that intriguing PT-induced toxicity profiles were recently reported [152], suggesting that research in radiobiology is still needed in PT. It also suggests that it could be interesting to investigate the mechanisms by which toxicities occur after PT, as they might differ slightly from the ones deriving from photons.

The physical properties of carbon ion beams and protons show a dosimetric advantage due to their characteristic dose deposit patterns. As linear energy transfer increases with the range in proton beams, RBE values can rise from about 1 in the entry channel to about 1.7 in the distal falloff region. In contrast, with carbon ions, the number and complexity of ionizations increase along with an increase in LET up to 100 keV per nucleon, which is the overkill area where death mechanisms are “saturated.” The resulting complex clustered DNA damage is more difficult to repair and can have a better therapeutic effect by killing tumor cells. However, low carbon ion doses could also induce complex unrepaired chromatid anomalies [153], which might contribute to second cancer events. More data on the mechanisms of repair and cell death are required for a better understanding of toxicities under different forms of radiotherapy. Sensitivity to uncertainties is higher with charged particles, the accuracy of IGRT [154], and the characterization of tissues and, in particular, of implanted materials [155].

Finally, proton therapy and ion therapy might be considered to reduce the volumes subjected to irradiation volumes—radiobiological and radiosensitivity investigations are ongoing.

Brachytherapy is also an effective way to reduce irradiated volumes using sources placed in the patient body. Currently, this is most often performed using a high dose rate due to radioprotection issues; it can also be performed as intracavitary or can be interstitial.

#### 4.2.2. Biological Mitigation

Hyperfractionation

Hyperfractionation is one modality of altered fractionation that has been historically used to reduce long-term toxicities. It delivers fraction doses of less than 1.8 Gy (often 1.2 Gy) more than once daily (leaving at least 6 h between two fractions to allow sufficient time for tissue repair) [156] or five times per week. This protective effect may differ in PT, as there might be a paradoxical deleterious effect of doses below 1.5 Gy RBE per fraction, such as those delivered with integrated boost irradiation schemes [157].

Hyper-radiosensitive patients are usually denied radiotherapy. Due to a lack of evidence-based data in moderately radiosensitive patients (group 2 in ATM studies) with molecularly detected iRS, no specific fractionation or technique recommendation has yet been proposed in radiosensitive patients [158]. While they should still undergo RT if RT is a mainstay of treatment for their cancer, a precautionary message may be to limit the irradiated volume and avoid severe hypofractionation. It may also be recommended not to include such patients in trials investigating dose-intense radiotherapy protocols. More generally, it would be interesting to have clinical trials dedicated to those patients.

Radiation effects including cell cycle arrest, DNA repair, and cell death) are well documented. However, specific gene expression data after different modalities of radiotherapy (conventional, hyper, or hypofractionated) are lacking implications for iRS testing and treatment adaptations could be important and clinically relevant.

New Frontiers: temporal fractionation (such as FLASH) and spatial fractionation

Another area of investigation in radiosensitive patients could be FLASH radiotherapy due to its dramatic normal tissue-sparing effects [159,160]. This disruptive ultra-high dose rate technology, currently developed using ultra-high dose rate electrons or protons, relies on the fact that the traversed normal tissues are normoxic or that radical oxygen species-induced peroxidation cascades differ between normal tissues and tumors. The FLASH effect is being investigated in early clinical trials [161,162]. More data on repair mechanisms, molecular interactions, and dose rates are warranted.

Other normal tissue-sparing technologies under development include high-dose spatial fractionation (GRID), allowing valleys of low doses along the beam paths [163]. 

Data specific to iRS are lacking. 

#### 4.2.3. Adapted Toxicity Management

Toxicities should be managed already from the early phase using appropriate supportive care to reduce the risk of them transforming into consequential late effects in some instances. Steroids or non-steroidal anti-inflammatory treatments—topical or systemic—are frequently used. RT may be continued, interrupted momentarily, or even permanently, depending on the severity of the effects. In addition to symptom-oriented supportive care, pre-rehabilitation and rehabilitation may improve long-term tolerance [164,165,166,167].

Late toxicity is generally considered irreversible, characterized by fibrosis and/or necrosis. When identified during follow-up, countermeasures can be proposed to mitigate these effects, such as:(1)Anti-inflammatory drugs or angiotensin II receptor antagonists [168]; corticosteroid therapy at a minimum dose of 1 mg/kg/day (equivalent prednisolone) is recommended for a minimum of 4 to 6 weeks.(2)Long-term use of antioxidant drugs such as superoxide dismutase or tocopherol (vitamin E), preferably combined with pentoxifylline, to mitigate fibrosis, as evidenced in randomized trials [169].(3)Bevacizumab, is an effective treatment of radionecrosis due to its intrinsic anti-edematous properties and preventive activity against anarchic angiogenesis, especially in the brain [170,171].(4)Endoscopic or minimally invasive procedures, may help to relief stenoses (e.g., esophagus, ureter), adhesions, and strictures (bowel). Argon plasma electrocoagulation of telangiectasia is an effective approach for skin or bleeding mucosae [165]. Several sessions are often necessary, especially for extended lesions. MRI-guided laser thermal ablation of brain radionecrosis was recently investigated.(5)Hyperbaric oxygen (HBO) is effective in better oxygenating fibrotic/hypoxic tissues, especially lymphedema, jaw osteoradionecrosis, and proctitis [166]. Clinical cases and open studies have reported a positive impact on bleeding after an average of 24 to 67 sessions. Some contraindications (claustrophobia, cardiac conduction disorders, uncontrolled epilepsy, bronchopathy, pneumothorax, etc.) may be observed.(6)In situ injection of autologous mesenchymal stem cells, recently advocated in the frame of accidental irradiation [167].

## 5. Conclusions

The understanding of increased intrinsic radiosensitivity in patients has substantially improved in the last decade. Despite possible morbidity and sequelae of “unusual” toxicities, systematic detection in patients with a normal phenotype has not yet been implemented in routine practice, mainly because proofs of performance of individual radiosensitivity testing have been available only recently. Other more complex reasons are the lack of financial support for this kind of monitoring and the lack of a proper assessment of the benefits of avoiding sequelae for the numerous patients who undergo radiotherapy at some point in their cancer history. Yet, it is challenging to estimate the risk/benefit ratio of radiotherapy in a given patient with a normal phenotype but with unusual toxicity for which iRS is suspected but not tested. iRS testing could contribute to a more clinically relevant decision. These gaps in knowledge in iRS testing should be addressed in order to design truly adapted radiotherapy protocols (technique, scheme, volumes, dose, combined modalities) and set up more frequent monitoring and more systematic supportive care.

## Figures and Tables

**Figure 1 cancers-14-06252-f001:**
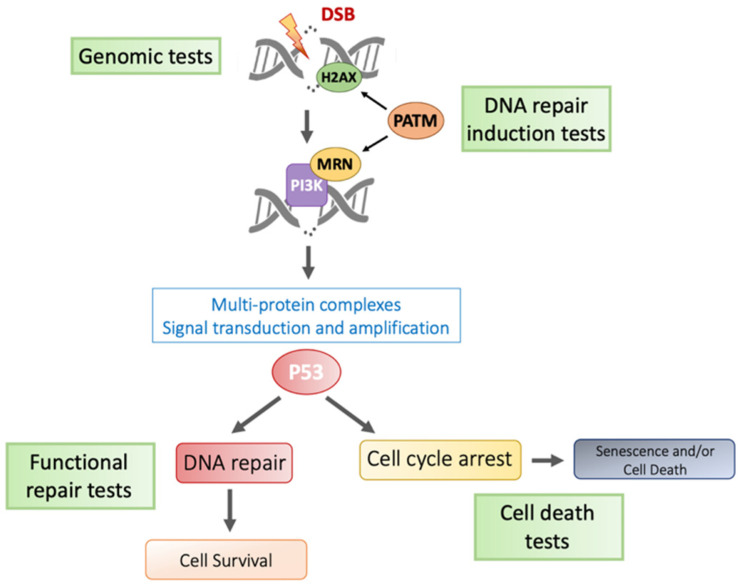
Main pathways involved in molecular toxicities and predictive assays.

**Figure 2 cancers-14-06252-f002:**
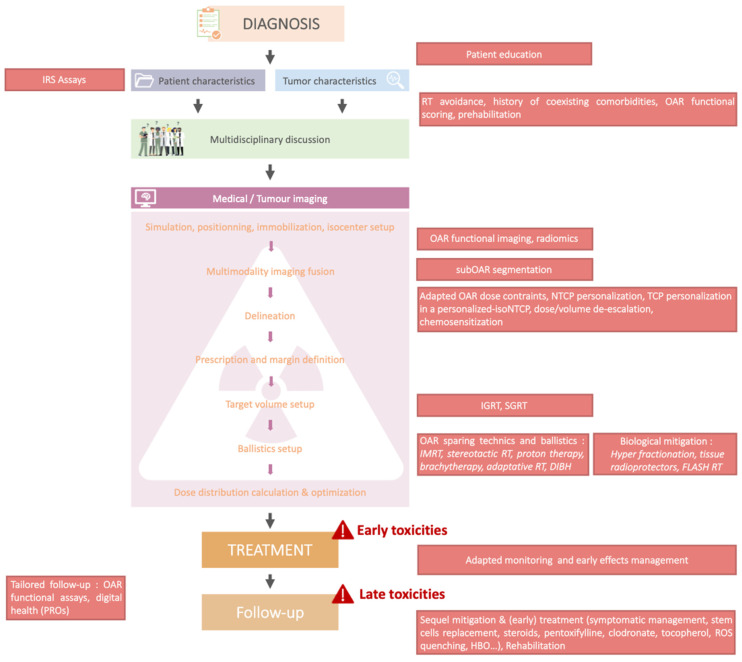
Practical consequences of knowing patients’ iRS of a patient before starting treatment.

**Table 1 cancers-14-06252-t001:** Main late toxicities observed in practice according to the tissue radiosensitivity, assuming the homogenous irradiation of the structures. OAR = Organ At Risk. Reprinted by permission from Springer Nature Customer Service Centre GmbH, Vogin, G. (2021) [9].

OAR	Main Late Toxicities
Radiosensitive OAR (Endpoint Occurring with 5-Year Occurrence Probability > 5% for Dose Usually < 20 Gy)	
Ovary	Infertility, premature ovarian failure, temporary or permanent castration
Testis	Temporary or permanent infertility
Lens	Cataract
Breast	Breast atrophy
Growth plates	Growth retardation or arrest
Kidney	Nephritis
Liver	Hepatitis, Radiation-Induced liver disease
Salivary glands	Temporary or permanent xerostomia
Bone marrow	±Deep/prolonged aplasia or selective blood cell loss (lymphocytes)
Mildly sensitive OAR (endpoint occurring with 5-year occurrence probability >5% for dose usually 20 Gy-60 Gy)	
Lung	Lung fibrosis—respiratory failure
Larynx	Dysphonia
Heart	Constrictive pericarditis, coronary artery stenosis, myocardial fibrosis, valvular damage
Small bowel	Enteritis, occlusive syndrome, perforation, fistula, malabsorption
Stomach	Late gastritis, ulceration, antral stenosis
Spinal cord	Late radiation myelitis
Hair	Depilation
Rectum	Late proctitis, ulceration, perforation, fistula
Bladder	Cystitis, micro bladder, ulceration, perforation, fistula
Brain—nerves	Necrosis, leukoencephalopathy, dementia, neurocognitive disorders, plexitis, neuropathy
Retina	Retinopathy, maculopathy
Thyroid	Hypothyroidism
Inner ear	Sensorineural deafness
Middle ear	Conductive deafness, chronic otitis media, eustachian tube pathology
Esophagus	Late esophagitis, ulcerations, fistulas
Mucosae	Mucositis, ulcerations, perforation, necrosis
Skin	Dystrophy, sclero-atrophic dermatitis, ulcerations
Radioresistant OAR (endpoint occurring with 5-year occurrence probability > 5% for dose usually > 60 Gy)	
Uterus—vagina	Endo-cervical-vaginal canal stenosis, uterine corpus fibrosis—infertility, vaginal synechiae, ulcerations, dryness, atrophy, vulvodynia
Bone	Osteoporosis, fracture, osteonecrosis
Muscles	Fibrosis
Joints	Ankylosis
Main arteries	Arterial disease, moya-moya vasculopathy
Connective tissues	Fibrosis

**Table 2 cancers-14-06252-t002:** Single nucleotide polymorphisms are linked with skin radiation-induced fibrosis.

Gene	Reference SNP	OR [CI95%]	Localization	Type of Mutation	References
ATM	rs1801516	1.23 [1, 1.51] 1.27 [1.02, 1.58]	Exonic	Missense (D > N)	[82,89]
ATM	IVS22–77 T>C	0.45 [0.24, 0.85]	Intronic	-	[86]
ATM	IVS48 + 238 C>G	0.50 [0.27, 0.94]	Intronic	-	[86]
DNMT1	rs2228611	0.26 [0.10, 0.71]	Exonic	Synonymous	[89]
TGFB1	rs1800469	3.40 [1.38, 8.40]	Upstream	-	[87,90]
TGFB1	rs1800470	2.37 [0.99, 5.60]	Exonic	Missense (P > L)	[87,90]
XRCC1	rs1799782 rs25487	4.33 [1.24, 15.12]	Exonic	Missense (R > W + Q > R)	[91]
XRCC3	rs861539	1.17 [1.09, 1.26]	Exonic	Missense (T > M)	[87]

**Table 3 cancers-14-06252-t003:** Transcriptomic studies comparing skin fibroblasts from radiosensitive and radioresistant patients.

Selected Differentially Expressed Genes	Assay Used for Gene Selection	References
FMLP-R-I, TNFα, NGFR, EPHB2, PDGFB, NTRK1, LFNG, DDR1; IFNGR1	Cytokine array	[90]
CDC6, CDON, CXCL12, FAP, FBLN2, LMNB2, LUM, MT1X, MXRA5, SLC1A3, SOD2, SOD3, WISP2	15KcDNA microarray	[93]
PLAGL1, CCND2, CDC6, DEGS1, CDON, CXCL12, MXRA5, LUM, MT1X, MT1F, MT1H, C1S, NF1, ARID5B, SCL1A3, TM4SF10, MGC33894, ZDHHC5/MFGE8	15K cDNA microarray	[94]
FBN2, FST, GPRC5B, NOTCH3, PLCB1, DPT, DDIT4L, SGCG	GeneChip Human Exon 1.0 ST Array	[95]

**Table 4 cancers-14-06252-t004:** Genome-wide association studies (GWAS) and significant polymorphisms associated with toxicities.

Type of Study	Tissue	External Validation (Yes/No)	Predictive Model (Yes/No)	Performance	References
SNP	Breast	Yes	Yes	OR = 4.47	[113]
				multivariate odds ratio	
		yes	No	[OR] = 0.77, *p* = 0.02	[82]
		No	Yes	OR = 1.78	[106]
		Yes	Yes	AUC = 0.65	[114]
	Lung	Yes	No	*p* = 0.031	[115]
		Yes	No	*p* = 0.02 to 0.023	[82,85]
	Prostate	No	No	OR = 1.3	[43,44,114]
		No	Yes	AUC from 0.76 to 0.8	
		Yes	Yes	AUC from 0.63 to 0.78	
GWAS	Prostate	No	Yes	OR from 6.42 to 33.95	[116]
		Yes	Yes	OR from 2.71 to 3.12	[117]
	Breast	Yes	No	RR from 1.56 to 3.28	[118]
		No	No	OR from 4.19 to 7.52	[119]

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
