# Peer review of "The Normal, the Radiosensitive, and the Ataxic in the Era of Precision Radiotherapy: A Narrative Review"

_cancers, 2022, doi:10.3390/cancers14246252_

Round 1

Reviewer 1 Report

The manuscript is well written. The longitudinal radiotherapy sensitivity effect follow-up studies novelty is supported by references. The review paves the way for more effective sensitisers and alternative cell signalling pathways inhibitions. 

Author Response

 The authors would particularly like to thank the reviewers who took time to read the article and to suggest corrections in our review. 

We tried to respond as best we can, to the different questions of the reviewers and hope that those details will be sufficient to clarify our point of view, in order to be accepted in your review. 

Below is the point-by-point response to reviewer's comments by color code: 

Bold : reviewers' comments 

Blue : authors’ responses 

Each modification was added to the text in yellow. 

Reviewer 2 Report

The review is comprehensive. Some minor issues need to be addressed.

1. Some reference could not show correctly(LINE 83, 297).

2. LET is linear energy transfer. (Line 252) 

3. When discussing heavy ions, why only helium ion is mentioned while current mainstay is carbon ion? (Line 604)

4. Brachytherapy is also photon-based technique. It may be misleading when putting it out of the section. (Line 606-608)

Author Response

 The authors would particularly like to thank the reviewer who took time to read the article and to suggest corrections in our review. 

We tried to respond as best we can, to the different questions of the reviewers and hope that those details will be sufficient to clarify our point of view, in order to be accepted in your review. 

Below is the point-by-point response to reviewer's comments by color code: 

Bold : reviewers' comments 

Each modification was added to the text in yellow. 

The entire article was proofread and corrected by an English fluent speaker. 

Reports:

1. Some reference could not show correctly (LINE 83, 297). 

Publications we were referring to were misplaced. This mistake was corrected 

2. LET is linear energy transfer. (Line 252) 

This typo was corrected 

3. When discussing heavy ions, why only helium ion is mentioned while current mainstay is carbon ion? (Line 604) 

We replaced heavy ions by ions beams in the text. Furthermore, it is indicated in line 605 that “However, low carbon ion doses could also induce complex unrepaired chromatid anomalies” to compare carbon ions effects to helium ions. 

4. Brachytherapy is also photon-based technique. It may be misleading when putting it out of the section. (Line 606-608) 

We added this information lines 653-654. 

Reviewer 3 Report

The manuscript turns deep insights into precision radiotherapy strategies. The review describes the methods of individual radiation sensitivity, their impact on toxicity management and possibility to create personalized radiotherapy protocols. Authors analyzed current knowledge of various tests in order to improve toxicity prediction. This manuscript is very interesting in guiding choice of radiotherapy and highlighting practical decisions.

Author Response

 The authors would particularly like to thank the reviewer who took time to read the article and to suggest corrections in our review. 

We tried to respond as best we can, to the different questions of the reviewers and hope that those details will be sufficient to clarify our point of view, in order to be accepted in your review. 

Reviewer 4 Report

This review article is well organized regarding to radiation therapy aspect and also well organized. I think with minor revision in English language is acceptable for this Journal.

Author Response

This review article is well organized regarding to radiation therapy aspect and also well organized. 

I think with minor revision in English language is acceptable for this Journal. 

We thank the reviewer for his report. English was improved in the manuscript.